# The Optimal Management of *Neisseria gonorrhoeae* Infections

**DOI:** 10.3390/microorganisms10122388

**Published:** 2022-12-01

**Authors:** Ramona Gabriela Ursu, Ionut Luchian, Costin Damian, Elena Porumb-Andrese, Roxana Gabriela Cobzaru, Giorgio Nichitean, Carmen Ripa, Diana Costin, Darius Sandu, Ioana-Maria Andrioaie, Luminita Smaranda Iancu

**Affiliations:** 1Department of Preventive Medicine and Interdisciplinarity (IX)—Microbiology, Faculty of Medicine, “Grigore T. Popa” University of Medicine and Pharmacy, 700115 Iasi, Romania; 2Department of Periodontology, Faculty of Dental Medicine, “Grigore T. Popa” University of Medicine and Pharmacy, 700115 Iasi, Romania; 3Department of Medical Specialties (III)-Dermatology, Faculty of Medicine, “Grigore T. Popa” University of Medicine and Pharmacy, 700115 Iasi, Romania; 4Faculty of Dental Medicine, “Grigore T. Popa” University of Medicine and Pharmacy, 700115 Iasi, Romania

**Keywords:** gonococcal infections, AMR detection assay, new drugs, NG vaccine candidate

## Abstract

*Neisseria gonorrhoeae* is one of the most frequent etiologic agents of STDs (sexually transmitted diseases). Untreated asymptomatic gonococcal infection in women can lead to spreading of the infection in the sexually active population and could lead to late consequences, such as sterility or ectopic pregnancies. One important issue about *N. gonorrhoeae* is its increasing resistance to antibiotics. This paper summarized the newest molecular antimicrobial resistance (AMR) detection assays for *Neisseria gonorrhoeae* connected with the latest therapeutic antimicrobials and gonococcal vaccine candidates. The assays used to detect AMR varied from the classical minimal inhibitory concentration (MIC) detection to whole-genome sequencing. New drugs against multi drug resistant (MDR) *N. gonorrhoeae* have been proposed and were evaluated in vivo and in vitro as being efficient in decreasing the *N. gonorrhoeae* burden. In addition, anti-*N. gonorrhoeae* vaccine candidates are being researched, which have been assessed by multiple techniques. With the efforts of many researchers who are studying the detection of antimicrobial resistance in this bacterium and identifying new drugs and new vaccine candidates against it, there is hope in reducing the gonorrhea burden worldwide.

## 1. Introduction

*Neisseria gonorrhoeae* is mentioned in the World Health Organization’s (WHO) priority pathogens list for research and development of new antibiotics, strains that are third-generation-cephalosporin-resistant and fluroquinolone-resistant are particularly mentioned [1]. The number of new cases is increasing in many countries, and the WHO has estimated more than 85 million new gonococcal infections in 2016 in for people aged 15–49 years [2]. *N. gonorrhoeae* prevalence and incidence is evaluated annually both by the CDC and ECDC [3,4], and the results are released as annual reports. It is known that asymptomatic, untreated gonorrhea in women can lead to spontaneous abortion, preterm birth, premature rupture of membranes, low birth weight and ophthalmia neonatorum [5]. The Centre for Genomic Pathogen Surveillance and the Euro-GASP study group underlined the importance of enhanced gonococcal AMR surveillance worldwide. The researchers suggested linking whole-genome sequence data to antimicrobial resistance data to fully understand the natural history of MDR (multi-drug resistant) *N. gonorrhoeae*. The authors also mentioned the need for new, efficient drugs to be developed [6]. Lin EY et al. analyzed data from the Pathogen Watch database and identified several combinations of genetic markers with high predictive values for decreased susceptibility to ceftriaxone, these being useful for developing new assays of antibiotic resistance detection [7]. 

In light of the recent data concerning the burden of *N. gonorrhoeae* antimicrobial resistance (AMR), in this paper we aimed to analyze the recent studies which tested *N. gonorrhoeae* AMR, studies which proposed new drugs for the therapy of gonorrhea and studies which presented new vaccine candidates against gonococci (Figure 1).

### 1.1. Molecular Assay Detection for AMR-Neisseria gonorrhoeae

*N. gonorrhoeae* AMR has been assessed in recent years with different assays, starting with classical methods, such as minimum inhibitory concentration (MIC) detection, together with modern methods, such as multilocus sequence typing (MLST), E-test and whole-genome sequencing (WGS), single-nucleotide polymorphisms in resistance-associated genes assays, WGS and bioinformatics analyses using the CGE, PATRIC and BLAST databases, RT PCR and high-resolution melting analysis for *porA* pseudogene detection, low-density hydrogel microarrays assays and Aptima Combo 2 assays.

The genes responsible for *N. gonorrhoeae* AMR are: *penA,* the most important factor affecting cephalosporin efficacy, *blaTEM-1B*, which is consistent with the antibiotic phenotype, the β-lactamase TEM genes and two novel TEM alleles, i.e., *rpsJ/V57M* and *tet(M)* mutations associated with tetracycline resistance, *gyrA_S91F* for ciprofloxacin, *16SrRNA_C1192U* for spectinomycin, *mtrR_G45D* for azithromycin and *penA_D545S* and *penA_mosaic* for cefixime/ceftriaxone resistance (Table 1).

### 1.2. New Therapeutic Antimicrobials against Neisseria gonorrhoeae

Given the actual increased prevalence of AMR for *Neisseria gonorrhoeae*, many researchers have tried to develop new possible treatments for this etiologic agent of STDs. A very recent paper from Australia presented an ionophore that disrupts metal homeostasis, namely PBT2. If PBT2 is administered simultaneously with zinc, this leads to AMR reversion for some etiological agents including *Neisseria gonorrhoeae*. The authors are considering PBT2 as a suitable therapy for MDR gonococcal infections [19]. Mullally C et al. presented small molecules that inhibit phosphor ethanol amine transferase A (EptA) to lipid A, and by using filter disc assays to evaluate the ability of different compounds to sensitize bacteria to polymyxin B, the authors showed that this favored the killing of *N. gonorrhoeae* by macrophages [20].

Researchers from the USA have tested first in vitro and then in vivo (in murine models) the activity of a gold-containing drug, auranofin, against the MDR *N*. *gonorrhoeae* strain FA1090. The authors found that auranofin reduced the vaginal burden of *N*. *gonorrhoeae* in comparison with the used control, and they concluded that auranofin could be considered as a suitable anti-*N*. *gonorrhoeae* therapy and should be further investigated [21]. Abutaleb NS et al. used also murine models to evaluate the efficiency of new drugs against *N*. *gonorrhoeae*, and they discovered that acetazolamide, a carbonic anhydrase inhibitor, repurposed as a novel inhibitor of *Neisseria gonorrhoeae*, was efficient in reducing the bacterial load in vaginal infections by 90% after 3 days of therapy [22]. A research team from Cambridge, United States, identified by sequencing a new therapeutic target, the class Ia ribonucleotide reductase (RNR) against MDR *N*. *gonorrhoeae,* and they demonstrated its efficiency in mice models [23] (Figure 2).

### 1.3. Feasibility of Gonococcal Vaccination

In the last two years, many researchers have tried to identify an optimal vaccine candidate against *N. gonorrhoeae*, and they have used different approaches. They used strains of gonococci and performed different comprehensive assays, they tested different adjuvants with optimal effect for vaccines, they used mouse female models and, very interestingly, they performed a bioinformatic analyses of more than five thousand gonococcal strains to identify suitable antigens for future vaccine candidates [24,25,26,27,28,29,30,31,32] (Table 2).

A very interesting finding is that outer membrane vesicle meningococcal serogroup B vaccines could have a protective effect against gonorrhea, this being mentioned by a few researchers, who have observed that vaccination against *Neisseria meningitidis* was associated with a reduced gonorrhea prevalence through a cross-protection mechanism against *Neisseria gonorrhoeae*. This observation could lead to the development of an effective and feasible gonococcal vaccine [33,34].

The feasibility of developing a gonococcal vaccine was presented by a report from a WHO global stakeholder consultation, with references to *N. gonorrhoeae* bacteriology and immune response and serogroup B meningococcal vaccines and gonorrhea vaccine development efforts [35] (Table 2). Two papers have so far been published in the Lancet journal that also support this finding that a meningococcal vaccine could be used to protect against *N. gonorrhoeae*, [36,37].

Researchers from Australia conducted a case-control study among infants, children and adolescents regarding the effectiveness and impact of the 4CMenB vaccine. The authors evaluated the vaccine’s effectiveness and impact on serogroup B meningococcal disease and gonorrhoea 2 years after the implementation of the programme, and they observed a sustained effectiveness against the serogroup B meningococcal disease 2 years after introduction in infants and adolescents and a moderate effectiveness against gonorrhoea in adolescents [37]. The authors of an international team mentioning the possibility of ending gonorrhea, as there is urgency in controlling this infection because of increasing antimicrobial resistance and the potential for it to become untreatable [36].

A very interesting paper from 2015, published in Scientific Reports, mentioned for the first time the possibility for an *N. gonorrhoeae* filamentous phage to induce antibodies with anti-gonococcal activity and that phage proteins may be a candidate for vaccine development. By DNA manipulation, transformation and phagemid particle preparation, the authors were able to present a vaccine candidate using a native bacteriophage as an immunogen, which could prevent disease sequelae such as PID [38].

The study of Kłyż A. et al. described the direct use of a filamentous phage as a potential vaccine against gonococci using the following comprehensive laboratory assays: the cultivation of bacterial strains, plasmids, phages, phage and phagemid particle preparation, electron microscopy, the production of polyclonal antisera, the determination of antibodies against phage particles by dot spot ELISA, the determination of the level of antibodies against *N. gonorrhoeae* cells by dot blot ELISA, a flow cytometric analysis, a serum bacterial assay and an adherence assay. The results of this research demonstrated that Ngo::Φ6fm can serve as an efficient antigen system and has the potential to form the basis for a vaccine against *N. gonorrhoeae* [39].

*N. gonorrhoeae* has been considered a versatile pathogen since 1987, when researchers from London mentioned the necessity of a better understanding of gonococcal pathogenicity, which could be useful for the development of an effective vaccine, as previous research on pili and outer membranes was not successful [40].

The issues with vaccine development have been presented in detail in a Nature review. The authors mentioned the mechanisms used by *N. gonorrhoeae* to evade the immune system, namely antigenic and phase variation in surface-exposed Type IV pili, Opa proteins and LOS in order to escape immunity, mimicking host molecules by some structure of the LOS and also mimicking human surface antigens, thus contributing to the difficulty of vaccine development [41].

Gulati S et al. mentioned that although gonococcal LOS-derived oligosaccharides (OSs) are major immune targets, they often undergo phase variation, a feature that seemingly makes LOS less desirable as a vaccine candidate. Thus, the author identified a gonococcal LOS-derived OS epitope called 2C7, and they considered that the 2C7 vaccine satisfied criteria that are desirable in a gonococcal vaccine candidate, namely a broad representation of the antigenic target, service as a virulence determinant, which is also critical for organism survival in vivo, and the elicitation of broadly cross-reactive IgG bactericidal antibodies when used as an immunogen [42].

Over the years, attempts to develop vaccines against *N. gonorrhoeae* based on different types of cell surface proteins and other molecules have ended without real success. In the last five years, four papers have been published regarding protein candidates for an effective vaccine against *N. gonorrhoeae*. Maurakis SA et al., from the USA, found that outer membrane TonB-dependent transporters (TdTs), which the microorganism employs to obtain metal ions from metal-binding proteins, are well conserved and essential in establishing infection, thus making them good candidates for protein-based vaccines [24]. Another team from the USA approached this problem by studying the most well-conserved antigens in over 5000 *N. gonorrhoeae* isolates and found that ACP, AniA, BamA, BamE, MtrE, NspA, NGO0778, NGO1251, NGO1985, OpcA, PldA, Slam2 and ZnuD had the highest degree of sequence conservation, and they found the *N. gonorrhoeae* FA1090 strain to be a good vaccine prototype as it carries antigenic sequences identical to the most broadly distributed types [27]. Sikora et al. assessed the potential of L-methionine-binding lipoprotein MetQ, located on the bacterial surface, to be used as a vaccine candidate [28]. The authors found that MetQ is remarkably well-conserved, with about 97% of isolates worldwide possessing the same variant, and they found that rMetQ combined with CpG as a Th1-stimulating adjuvant induced protective immunity in mice [28]. In another mice model study by Almonacid-Mendoza HL et al., the authors examined the possibility of using recombinant the *Neisseria gonorrhoeae* Adhesin Complex Protein (rNg-ACP) in a vaccine and found that it induced the production of bactericidal antibodies that also stopped the gonococcus from inhibiting innate lytic defense molecules [43].

## 2. Discussion and Conclusions

*N. gonorrhoeae* infection still represents an important public health issue due to multiple factors that contribute to its virulence and worldwide spread, such as its extraordinary capacity to develop resistance to all antibiotics introduced as standard treatment, the large number of asymptomatic infections, which constitute a reservoir, cultural stigma associated with STDs in some regions of the world and the lack of efficient prevention. The latest studies performed on all continents identified AMR for NG in different percentages: more than 16% for ceftriaxone in China [8], 88% for ciprofloxacin, tetracycline and benzylpenicillin in Sweden [9], 48.2% for ciprofloxacin in Spain [14], 89% for tetracycline in South Africa [15], 99.6% for ciprofloxacin in Uganda [17] and 86.8% for ciprofloxacin in Australia [18]. Even though the reported AMR was high, a homogenous conclusion cannot be drawn, as the authors all used totally different assays for testing AMR, each with different values of sensitivity, specificity, positive predictive values and negative predictive values. For example, we could not compare MIC detection offered by an E-test with resistance genes detected by WGS.

Using the model of other multidrug-resistant strains, it could be useful to systematically assess the long-term clearance efficacy of drug combinations by quantifying gonococcal survival under antibiotic exposure for developing more-effective, resistance-proof multidrug regimes [44].

Similar to other STDs, for example HPV (human papilloma virus) [45], the standardization of testing for *Neisseria gonorrhoeae* is of paramount importance. Ideally, an assay with optimal test parameters needs to be developed, and it has to be at an affordable price for all countries. In this way, implementing prevention strategies at a country level, but under WHO guidance, would be possible.

The optimal management of gonococcal infection involves synergistic actions, namely between the detection of AMR and identifying new therapeutic drugs and candidate vaccines against *N. gonorrhoeae.* For example, ECDC organized, in 2019, a gonococcal antimicrobial susceptibility surveillance, which supposed that antimicrobial susceptibility testing was performed using MIC (minimal inhibitory concentration) gradient strip tests, which used mainly E-tests, for gonococcal strains isolated from genital, pharyngeal and anorectal samples. The bacterial strains were tested for ceftriaxone, cefixime, azithromycin and ciprofloxacin, including the production of penicillinase, resulting in high-level penicillin resistance with nitrocefin. This surveillance program detected a decreasing trend in the levels of cefixime resistance in 2019 and a continued rise in both ciprofloxacin resistance and azithromycin MICs along with the detection of three ceftriaxone-resistant isolates [46].

Similar data regarding gonococcal resistance has been reported by Cole MJ et al. and Unemo M et al. The Collaborative Clinical Group Gonorrhoea Survey included some suggestions for the significant enhancement of gonococcal infection control, namely the testing of cures, contact tracing for patients with confirmed gonorrhoea, access to validated and quality-assured cultures, AMR testing, nucleic acid amplification tests (including the confirmation of gonococcal reactive samples) and a recommended antimicrobial treatment [47,48].

The update to the CDC’s therapy guidelines for gonococcal infection from 2020 contains information regarding the need for antimicrobial stewardship and the potential impact of the appropriate use of dual therapy on the microbiome and on pathogenic organisms [49].

Gonococcal infection is a real public health concern, and the British Association for Sexual Health and HIV also referred to antimicrobial stewardship, which could be optimized for gonococcal infection, but could mainly be used for negative gonorrhoea contacts, these being tested through sensitive assays [50]. Belgium is another European country which has developed its own guidelines for gonorrhoea infection. This was developed after Belgian doctors noticed non-adapted suboptimal care when using only international guidance [51]. Using the model of other multidrug resistant strains, it would be useful to systematically assess the long-term clearance efficacy of drug combinations by quantifying gonococcal survival under antibiotic exposure in order to develop more-effective, resistance-proof multidrug regimes [44].

Another good initiative taken by the ECDC in optimizing the gonococcal infection control was organized in 2019, which took the form of an external quality assessment laboratory-based surveillance system. This system was intended to compare the performance of each participating laboratory for performing antimicrobial susceptibility testing in Neisseria gonorrhoeae. The laboratories participating in this surveillance showed good levels of competency and capability in testing strains of unknown phenotypes [52].

A recent ECDC document from September 2022 mentioned the initiative of analysing the genome data of *Neisseria gonorrhoeae* strains together with epidemiological data in order to gather information on the geographical and temporal distribution patterns of public-health-relevant strains in the EU/EEA region. The ECDC intend to collect data related to gender, country of birth, age, pathogen genome sequence, associations between genotype, antimicrobial resistance and patient characteristics [53]. The fact that data will be collected anonymously, as personal data is not mandatory, could help to better control this infection [53].

The main action for preventing and controlling antimicrobial resistance are the prudent use of antimicrobials (only when needed, correct dose, dose intervals and duration), infection control (hand hygiene, screening and isolation) and developing new antibiotics (with a novel mechanism of action, research and development) [54,55].

The new method of altering the action of the enzyme’s quaternary structure by PTC-847 and PTC-672, two novel orally active inhibitors with a narrow spectrum of activity against *N. gonorrhoeae,* is just one example of sustaining the optimization of the management of gonococcal infection as part of antimicrobial stewardship [23].

With a standardized screening procedure, with new drugs being developed and validated in clinical studies and with the efforts to develop an efficient vaccine against *N. gonorrhoeae*, there is hope for ending this disease.

## Figures and Tables

**Figure 1 microorganisms-10-02388-f001:**
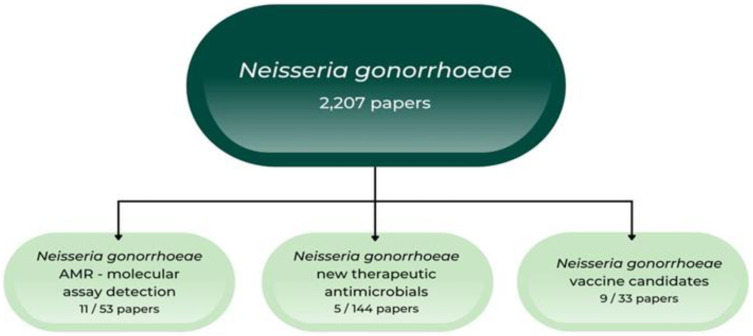
Diagram of analyzed studies.

**Figure 2 microorganisms-10-02388-f002:**
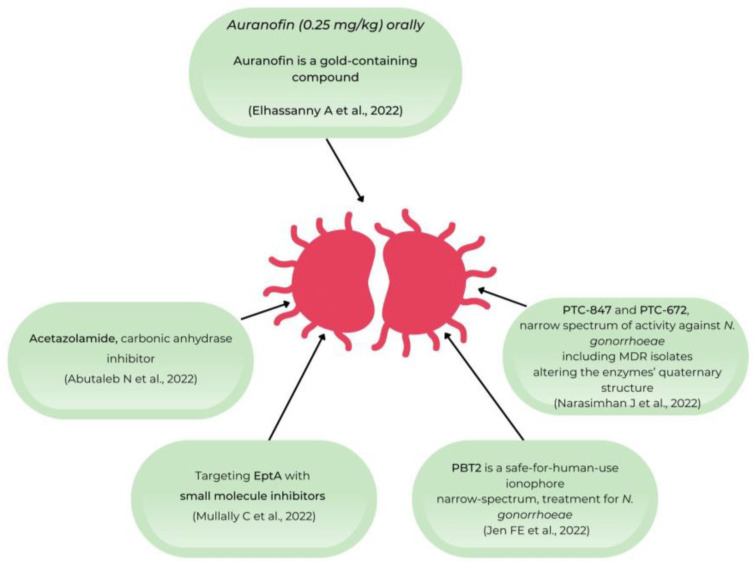
Schematic representation of the new therapeutical drugs against *N. gonorrhoeae* [19,20,21,22,23].

**Table 1 microorganisms-10-02388-t001:** AMR detection assays for *Neisseria gonorrhoeae*.

Authors Year, Country	Sample Type/Objective	AMR Assay Detection	Results	Novelty
Lin X. et al., 2022, Guangdong, People’s Republic of China [8]	4113 isolates To clarify the AMR trend from 2016–2019 and reveal the molecular characteristics and evolution of ceftriaxone-resistant penA 60.001 isolates.	The MIC determination was conducted using the agar dilution method. *N. gonorrhoeae* multiantigen sequence typing (NG-MAST), multilocus sequence typing (MLST) and *N. gonorrhoeae* sequence typing for antimicrobial resistance (NG-STAR) were used to identify the sequence types.	Isolates with decreased susceptibility were detected. Six ceftriaxone-resistant isolates possessing *penA* 60.001 appeared in Guangdong Province and were resistant to ceftriaxone, penicillin, tetracycline, ciprofloxacin and cefixime, but they were susceptible to azithromycin and spectinomycin.	A set of virulence factors were detected by genome analysis along with the resistance genes “*penA*” and “*blaTEM-1B*”. Half of the penA 60.001 strains were fully mixed with global FC428-related strains. Global FC428-related clones have disseminated across Guangdong, possibly causing decreased ceftriaxone susceptibility. Enhanced gonococcal surveillance measures will help elucidate the trajectory of transmission and curb further dissemination.
Golparian D et al., 2022, Örebro, Sweden [9]	To investigate the genomic diversity, AMR and AMR determinants in gonococcal isolates cultured in 2018 in Bangkok, Thailand.	Gonococcal isolates from males (n = 37) and females (n = 62) were examined by an E-test and whole-genome sequencing (WGS). AMR determinants and molecular epidemiological STs were characterized.	A remarkably high frequency (88%) of *β-lactamase TEM genes* was detected, and two novel TEM alleles were discovered.	The high prevalence of AMR and AMR determinants for ciprofloxacin, tetracycline and benzylpenicillin, plus some strains belonging to clones/clades especially in sublineage A2 that were prone to develop resistance to extended-spectrum cephalosporins (ESCs), such as ceftriaxone, and azithromycin, showed the importance of continued and strengthened AMR surveillance, including WGS, of *N. gonorrhoeae* in Thailand.
Li Y et al., 2022, Beijing, China [10]	To develop a high-throughput multiplex assay that incorporates high-resolution melting technology based on a six-codon assay.	The method develop by this team could precisely distinguish specific single-nucleotide polymorphisms in resistance-associated genes with a specificity and sensitivity of 100% and a detection limit as low as 10 copies per reaction.	The method could be applied directly to clinical samples without the need for cultivation and it successfully predicted all cephalosporin-resistant isolates (sensitivity: 100%).	This technique allowed for the rapid testing of antimicrobial resistance and will serve as a valuable tool for tailored antimicrobial therapy and for monitoring the transmission of cephalosporin-resistant strains.
Cassu-Corsi D et al., 2022, São Paulo, SP, Brazil [11]	Thirteen *Neisseria gonorrhoeae* isolates were sequenced, and they exhibited distinct susceptibility profiles. The samples were collected over 12 years in the metropolitan region of São Paulo, Brazil.	WGS was carried out on Illumina MiSeq™ 2 × 300 bp paired-end reads. Bioinformatics analyses were performed using CGE, PATRIC and BLAST databases for manual curation of the genomes obtained.	Tetracycline resistance determinants, namely *rpsJ/V57M* and *tet(M)*, were detected, as well as mutations associated with sulfonamides and rifampicin resistance.	These results showed the patterns of AMR in *N. gonorrhoeae* strains that circulated in Brazil.
Mitchev N et al., 2022, Durban, South Africa [12]	The research team wanted to detect single-nucleotide polymorphisms and plasmids associated with AMR from *N. gonorrhoeae* isolates from a population in South Africa.	Real-time PCR was used as a rapid test for AMR detection. RT PCR and high-resolution melting analysis were used to detect the porA pseudogene (species-specific marker) and resistance-associated targets.	Concordance was found between molecular detection (real-time PCR and HRM), and the resistance phenotype was ≥92% for *blaTEM* (HLR penicillin), *rpsJ_V57M* (tetracycline), tetM (tetracycline) and gyrA_S91F (ciprofloxacin). The resistance determinants *16SrRNA_C1192U* (spectinomycin), *mtrR_G45D* (azithromycin) and *penA_D545S* and penA_*mosaic* (cefixime/ceftriaxone) were correlated with the WHO control isolates.	Eight resistance-associated targets correlated with phenotypic culture results. The porA pseudogene reliably detected *N. gonorrhoeae*. Larger cohorts are required to validate the utility of these targets as a convenient, fast diagnostic tool without the need for cultivation to guide STI management in a South African population.
Shaskolskiy B et al., 2022, Moscow, Russia [13]	A total of 448 clinical isolates were evaluated for determinants of ceftriaxone resistance. The samples, which were collected in Russia, were identified using microarray analysis, and MIC values for ceftriaxone (MICcro) were calculated using the regression equation and compared with those measured by the serial dilution method.	A method for predicting the *N. gonorrhoeae* ceftriaxone susceptibility level was conducted by identifying genetic determinants of resistance using low-density hydrogel microarrays and a regression equation.	No ceftriaxone-resistant isolates were identified in the analyzed samples, and no interpretative errors were detected in the MICcro calculations.	The developed strategy for predicting ceftriaxone MIC can be used for the continuous surveillance of known and emerging resistant *N. gonorrhoeae* isolates.
Alonso R et al., 2021, Vitoria-Gasteiz, Spain [14]	The aim of this study was to apply molecular epidemiology, antimicrobial surveillance and PK/PD analysis to guide the antimicrobial treatment of gonococci infections in a region of the north of Spain.	Thirty-five isolates that were intermediate or resistant to at least two antimicrobials were selected to search for resistance genes and genotyping through whole-genome sequencing (WGS).	In total, 2.0%, 6.4%, 5.4% and 48.2% of the isolates were resistant to ceftriaxone, cefixime, azithromycin and ciprofloxacin, respectively.	We can confirm that ceftriaxone (even at the lowest dose: 250 mg) and oral cefixime were good candidates to treat gonorrhea. For patients allergic to cephalosporins, ciprofloxacin should be only used if the MIC is known and ≤0.125 mg/L; this antimicrobial is not recommended for empirical treatment.
Mitchev N et al., 2021, Durban, South Africa [15]	This study investigated the molecular epidemiology and AMR profiles of *Neisseria gonorrhoeae* isolates in KwaZulu-Natal province (KZN), South Africa.	An E-test was performed to determine antimicrobial susceptibility. WGS was used to determine epidemiology and to predict susceptibility by detecting resistance-associated genes and mutations.	Two novel *penA* alleles and eight novel *mtrR* alleles were identified.	This study revealed a high prevalence of AMR (penicillin 67%, tetracycline 89% and ciprofloxacin 52%). However, in the case of these strains, spectinomycin, cefixime, ceftriaxone and azithromycin remained 100% effective. The emergence of resistance to cefixime, ceftriaxone and azithromycin is still possible and continued surveillance is crucial to monitor it.
Reimche JL et al., 2018, Centers for Disease Control and Prevention, Atlanta, GA [16]	Genomic analysis provided a powerful tool for the surveillance of circulating strains, antimicrobial resistance determinants and an understanding of transmission through a population.	Whole-genome sequencing was used. This study provided a WGS data set from a systematically collected and well-characterized sample of isolates for the analysis and prediction of AMR profiles from diverse genomic backgrounds. Although WGS cannot replace phenotypic testing, routine molecular surveillance is crucial for detecting the accumulation of or change in AMR variants within a short period and for monitoring the trends of select strains harboring AMR markers. This knowledge will guide the development of novel diagnostics and personalized treatments and will ultimately direct strategies for the mitigation of the emergence and spread of resistant GC strains in the population.	Common antimicrobial resistance determinants were associated with low-level or high-level decreased susceptibility or resistance to relevant antibiotics.	The characterization of this 2018 Gonococcal Isolate Surveillance Project genomic data set, which was the largest US whole-genome sequence data set to date, set the basis for future prospective studies and established a genomic baseline of GC populations for local and national monitoring.
Kakooza F et al., 2021, Kampala, Uganda [17]	The study aimed to describe the establishment, design and implementation of a standardized and quality-assured gonococcal surveillance program and to describe the susceptibility patterns of the cultured gonococcal isolates in Kampala, Uganda.	Males completed a questionnaire and provided a urethral swab specimen. Culture, identification and antimicrobial susceptibility testing (E-test) were performed.	Of the 1013 males recruited, 73.1% (740/1013) had a positive Gram stain and 51.1% (n = 518) were culture-positive for *Neisseria gonorrhoeae*. Using an E-test (458 isolates), the resistance to ciprofloxacin was 99.6%. Most isolates were susceptible to azithromycin, cefoxitin and gentamicin, that is, 99.8%, 98.5% and 92.4%, respectively, and all isolates were susceptible to ceftriaxone and cefixime.	These isolates with epidemiological data can be used to develop population-level interventions for the surveillance and control of this disease.
Lee DYJ et al., 2021, Victoria, Australia [18]	A ResistancePlus GC assay demonstrated a 94.8% sensitivity and 100.0% specificity for *Neisseria gonorrhoeae* detection.	Aptima Combo 2 assay.	Of the 379 concordant *N. gonorrhoeae*-positive samples, in 86.8% of cases the *gyrA S91F* mutation was found, which was highly predictive for ciprofloxacin resistance	This study supported the feasibility of implementing RGT for gonorrhea into routine molecular workflows.

**Table 2 microorganisms-10-02388-t002:** Gonococcal vaccine candidates.

Authors Year, Country	Hypothesis/Objective	Material and Methods	Results	Novelty
Maurakis SA et al., 2022, Atlanta, Georgia, USA [24]	- TdfJ loop 3 helix (L3H) plays a similar role in the binding of and zinc extraction from S100A7. - The affinity of wild-type (WT) TdfJ and S100A7 was determined, and then, to identify key amino acid residues of TdfJ involved in binding and subsequent zinc extraction from S100A7, site-directed mutagenesis was utilized.	- Strains of *N. gonorrhoeae* were cultivated and maintained on GC medium base (Difco) agar with Kellogg’s supplement I. Genetic analysis required gonococcal mutant construction, expression plasmid construction and recombinant protein purification. - Surface plasmon resonance. - Whole-cell dot blots. - Purified protein ELISAs. - Zinc-restricted growth assay. - Zinc internalization assay. - Western blotting.	- Wild-type TdfJ bound S100A7 with a high affinity. - Gonococci expressing mutated tdfJ were defective for S100A7 utilization. - TdTs continue to feature as promising targets for vaccine and therapeutic development to combat gonococcal disease.	- This was the first time the binding interaction for gonococcal TdfJ and its human ligand S100A7 had been reported. - Several mutations in TdfJ loop 3 that alter S100A7 binding and subsequent zinc extraction were identified. - There is a need for understanding of their virulence mechanisms, and a similar characterization of the other TdT-ligand pairs will hopefully follow.
Joshi D et al., 2021, Atlanta, Georgia, USA [25]	This study aimed to identify potential vaccine adjuvants, which, according with their mechanism of action, could be: immunostimulatory adjuvants, vaccine delivery systems.	Microparticulate formulation of (S)-DPD (-4,5-Dihydroxy-2,3-pentanedione) immunogenicity and adjuvant potential was tested via in vitro evaluation.	Microparticulate (S)-DPD was found to be non-cytotoxic towards the antigen-presenting cells and had an adjuvant effect with the microparticulate gonorrhea vaccine.	Further studies with additional bacterial vaccines and an in vivo evaluation are required to confirm the potential of microparticulate (S)-DPD as a probable vaccine-adjuvant candidate.
Jiao H et al., 2021, Yangzhou, China [26]	A novel *Neisseria gonorrhoeae* DNA vaccine delivered by *Salmonella* Enteritidis (SE) ghosts was developed, and the immune responses of the vaccine candidate were evaluated.	*Neisseria gonorrhoeae* nspA gene was cloned into the pVAX1 vector. The constructed recombinant plasmid pVAX1-nspA was loaded into lyophilized *Salmonella* Enteritidis ghosts to produce SE ghosts (pVAX1-nspA).	The co-administration of SE ghosts (pVAX1-nspA) and SE ghosts (pVAX1-porB) had the highest bactericidal antibody titers.	This study demonstrated the potential of the co-administration of SE ghosts (pVAX1-nspA) and SE ghosts (pVAX1-porB) as an attractive vaccination regimen for gonorrhea.
Baarda BI et al., 2021, Oregon, USA [27]	The remarkable variation in surface exposed antigens of *N. gonorrhoeae* has been a longstanding barrier in developing an effective vaccine.	Bioinformatic analyses of sequence variation were conducted by comparing 34 gonorrhea antigen candidates among >5000 clinical *N. gonorrhoeae* isolates deposited in the Neisseria PubMLST database.	ACP, AniA, BamA, BamE, MtrE, NspA, NGO0778, NGO1251, NGO1985, OpcA, PldA, Slam2 and ZnuD were found to have a high degree of sequence conservation and exhibited the distribution of a single antigen variant among *N. gonorrhoeae* strains globally or via low-frequency polymorphisms in surface loops, which make them promising candidates for a gonorrhea vaccine.	The commonly used *N. gonorrhoeae* FA1090 strain emerges as a vaccine prototype, as it carries antigen sequence types identical to the most broadly distributed antigen variants. Extensive bioinformatic analyses were used in trying to solve the problem of finding suitable antigen candidates.
Sikora AE et al., 2020, Oregon, United States [28]	L-methionine-binding lipoprotein MetQ (NGO2139) and its homolog GNA1946 (NMB1946) were identified by reverse vaccinology as gonococcal and meningococcal vaccine candidates, respectively.	The suitability of MetQ for inclusion in a gonorrhea vaccine was studied by examining MetQ conservation, its function in *Neisseria gonorrhoeae* pathogenesis and its ability to induce protective immune responses using a female murine model of lower genital tract infection. - The study involved mice immunized with rMetQ-CpG (n = 40), a vaccine containing a tag-free version of MetQ f.	rMetQ-CpG-immunized mice cleared infection faster than control animals and had a lower *N. gonorrhoeae* burden.	rMetQ-CpG induced a protective immune response that accelerated bacterial clearance from the murine lower genital tract and represents an attractive component of a gonorrhea subunit vaccine.
Holley CL et al., 2020, Atlanta, USA [29]	This study aimed to define the mechanism of the MisR regulation of ompA, thus providing a model for the MisR activation of this *N. gonorrhoeae* virulence gene.	- Bacterial strains, plasmids and primers. - The generation of ompA and misR-null mutants. - Complementation of the ompA::ermC mutant. - qRT-PCR analysis of Ng transcripts. - Purification of recombinant His-OmpA protein and preparation of polyclonal antisera. - Western blotting. - EMSA for the detection of MisR binding to target DNA. - Primer extension analysis. - Construction of the ompA-lacZ fusions. - Preparation of cell extracts and β-galactosidase assay.	This was the first report that has characterized the regulation of the *N. gonorrhoeae* ompA gene, which encodes a candidate vaccine antigen.	- MisR/MisS directly enhances ompA expression. - The intrinsic linkage of MisR/MisS and OmpA could be exploited for vaccine or chemotherapeutic development purposes.
Kammerman MT et al., 2020, Atlanta, Georgia, USA [30]	This study was motivated by the increased interest in identifying new bacterial targets for the treatment and, ideally, prevention of gonococcal disease. TdfH (TonB-dependent transporter) is highly conserved among the pathogenic *Neisseria* species, making it a potentially promising candidate for inclusion into a gonococcal vaccine.	Neisserial growth conditions. Cloning, expression and purification of TdfH. hCP, hTf, bTf, preparation and metal loading. CP (calprotectin)-dependent growth of *N. gonorrhoeae*. Isothermal titration calorimetry (ITC) Whole-cell dot blot competition assays and total calprotectin binding assays. TdfH-CP complex generation and characterization. Alignment of human and mouse S100A8 and S100A9 protein sequence. Homology modeling of TdfH.	TdfH specifically interacted with human calprotectin (hCP) and the growth of the gonococcus was supported in a TdfH-dependent manner only when hCP was available as a sole zinc source and not when mouse CP was provided.	An antigonococcal therapeutic could potentially block this site on calprotectin, interrupting Zn uptake by *N. gonorrhoeae* and thus stopping bacterial growth. TdfH and calprotectin protein–protein interactions were described, and these findings provide the building blocks for future therapeutic or prophylactic targets.
Semchenko EA et al., 2020, Gold Coast, Australia [31]	This research team evaluated the gonococcal *Neisseria*-heparin-binding antigen (NHBA) as a potential vaccine candidate in terms of its sequence conservation and expression in a range of *N. gonorrhoeae* strains. In addition, its immunogenicity and the functional activity of its antibodies raised to either the full length NHBA or a C-terminal fragment of NHBA (NHBA-c) were studied.	- Bacterial strains and growth conditions. - Sequence analysis. - Construction of *N. gonorrhoeae* NHBA mutant strains. - Recombinant protein expression. - Generation of polyclonal antibodies. - Enzyme-linked immunosorbent assays (ELISAs). - Serum bactericidal activity (SBA) and opsonophagocytic killing (OPA) assays. - Flow cytometry analysis, - Surface plasmon resonance (SPR). - Epithelial cell adherence assays.	- Recombinant NHBA had immunogenic potential. - Mice immunized with either NHBA or NHBA-c adjuvanted with either Freund’s or aluminium hydroxide exhibited a humoral immune response with predominantly IgG1 antibodies.	Anti-NHBA was also able to inhibit the functional activity of NHBA by reducing binding to heparin and adherence to epithelial cells of the cervix and the urethra. The gonococcal NHBA is a promising vaccine antigen to include in a vaccine to control *N. gonorrhoeae*.
Gulati S et al., 2019, Massachusetts, USA [32]	The 2C7 mimitope was evaluated as a potential candidate for a vaccine. It has since been configured as a stable, highly pure tetrapeptide and administered with glucopyranosyl lipid A (GLA) formulated in a stable oil-in-water nanoemulsion (SE) as an adjuvant approved for use in humans.	- Bacterial strains. - Materials for the production of a TMCP2 vaccine candidate. - Immunization of mice. - ELISA to measure anti-LOS antibody levels. - Inhibition ELISA. - Depletion of mouse IgM. - Serum bactericidal assays. - Mouse protection experiments.	The candidate gonococcal peptide vaccine that elicits bactericidal antibodies against *N. gonorrhoeae * significantly reduces the duration and burden of gonococcal cervicovaginal colonization in BALB/c mice.	TMCP2 represents an important step forward in the development of a safe, economical and effective gonococcal vaccine.

## Data Availability

Not applicable.

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
