# Peer review of "The Optimal Management of Neisseria gonorrhoeae Infections"

_microorganisms, 2022, doi:10.3390/microorganisms10122388_

Round 1

Reviewer 1 Report

I have few suggestions to improve this review article:

Line 24, we identified different assay used to detect AMR. This sentence makes sense that authors of this review article developed methods.

Is it know that treatment of Neisseria gonorrhoeae with single antibiotic vs combinations of antibiotic and its output. This study could be a good reference:

https://www.nature.com/articles/s41586-022-05260-5

Having figure for New therapeutic antimicrobials against Neisseria gonorrhoeae with targets will be catchy and easy to understand. Pictorial representation is always great.

Line 110, In regards of this find, that a meningococcal vaccine could be…… this sentence is not correct. Please correct sentences with flow.

Line 111, two papers have so far been published in the Lancet journal. What are the take home message of these two papers should be briefly discussed for clarity.

Author Response

Point-by-point response to the reviewers’ and editor’s comments

Title: “The Optimal Management of Neisseria gonorrhoeae Infections”

We thank the reviewer for giving us the opportunity to improve the quality of our manuscript.

Reviewer #1:

Comments and Suggestions for Authors

I have few suggestions to improve this review article:

  1. Line 24, we identified different assay used to detect AMR. This sentence makes sense that authors of this review article developed methods.
    • Response to Reviewer: We thank the reviewer for this comment. We modified the sentence as suggested (line 24 – 25).

The assays used to detect AMR varied from the classical MIC detection to whole genome sequencing.

  1. Is it know that treatment of Neisseria gonorrhoeae with single antibiotic vs combinations of antibiotic and its output. This study could be a good reference:

https://www.nature.com/articles/s41586-022-05260-5

  • Response to Reviewer: We thank the reviewer for this comment. We introduced the suggested reference, and we added the date from this paper in Discussion data (lines 201 - 204).

Using the model of other multi drug resistant strains, could be useful to systematically assess the long-term clearance efficacy of drug combinations by quantifying gonococcal survival under antibiotic exposure, for developing more-effective, resistance-proof multidrug regimes (Lázár V et al., 2022).

Lázár V, Snitser O, Barkan D, Kishony R. Antibiotic combinations reduce Staphylococcus aureus clearance. Nature. 2022 Oct 5:1–7. doi: 10.1038/s41586-022-05260-5. Epub ahead of print. PMID: 36198788; PMCID: PMC9533972.

  1. Having figure for New therapeutic antimicrobials against Neisseria gonorrhoeae with targets will be catchy and easy to understand. Pictorial representation is always great.
    • Response to Reviewer: We thank the reviewer for this comment. We introduced the suggested pictorial representation as figure 2 (lines 86 – 88):

Figure 2: Schematic representation on the new therapeutical drugs against N. gonorrhoeae

  1. Line 110, In regards of this find, that a meningococcal vaccine could be…… this sentence is not correct. Please correct sentences with flow.
    • Response to Reviewer: We thank the reviewer for this comment. We modified the sentence as suggested (lines 119 – 121).

Supporting this finding, that a meningococcal vaccine could be used to protect against N. gonorrhoeae as well, two papers have so far been published in the Lancet journal

  1. Line 111, two papers have so far been published in the Lancet journal. What are the take home message of these two papers should be briefly discussed for clarity.
    • Response to Reviewer: We thank the reviewer for this comment. We add more data from these two papers, as suggested (line 122 – 131).

Researchers from Australia conducted a case-control study among infants, children and adolescents regarding the effectiveness and impact of the 4CMenB vaccine. The authors evaluated vaccine effectiveness and impact on serogroup B meningococcal disease and gonorrhoea 2 years after implementation of the programme, and they observed sustained effectiveness against serogroup B meningococcal disease 2 years after introduction in infants and adolescents, and moderate effectiveness against gonorrhoea in adolescents (Wang, B et al., 2022). The authors of an international team are mentioning of the possibility of ending gonorrhea, as there is urgency in controlling this infection because of the increasing antimicrobial resistance and potential to become untreatable (Ong, J.J et al., 2022).

  1. Ong, J.J.; Unemo, M.; Choong, A.L.; Zhao, V.; Chow, E.P. Is the End of Gonorrhoea in Sight? Lancet Infect Dis 2022, 22, 919–921, doi:10.1016/S1473-3099(22)00002-0.
  2. Wang, B.; Giles, L.; Andraweera, P.; McMillan, M.; Almond, S.; Beazley, R.; Mitchell, J.; Lally, N.; Ahoure, M.; Denehy, E.; et al. Effectiveness and Impact of the 4CMenB Vaccine against Invasive Serogroup B Meningococcal Disease and Gonorrhoea in an Infant, Child, and Adolescent Programme: An Observational Cohort and Case-Control Study. Lancet Infect Dis 2022, 22, 1011–1020, doi:10.1016/S1473-3099(21)00754-4.

Reviewer 2 Report

The title not completely corresponds to content of the article. What do you mean by “Optimal Management”?

The English language and grammar should be carefully checked. Numerous errors are present, the language should be corrected

As for the content of the article, it should be noted that the article is written sloppily. There is a lot of mistakes. I noted the most bizarre-looking ones.

1)       Figure 1 – two sub-blocks with the same text inside (Neisseria gonorrhoeae AmR- molecular assay detection 11/53 papers). Please, correct.

2)       Lines 64-69. The list of genes responsible for AMR resistance should be checked:

(a) blaTEM-1 is not associated with ceftriaxone resistance.

(b) There is no β-lactamase TEM genes associated with ciprofloxacin, tetracycline and benzylpenicillin resistance.

3) Сheck the references.

Author Response

Thank you !

Reviewer 3 Report

The manuscript needs to be completed with the information shown below.

·         Why were not all publications included in the review (Figure 1)? What was the criterion for selecting publications?

·         Please complete the information on vaccine candidates based on filamentous phages;

·         There is also a lack of information on the reasons for the difficulty in obtaining an effective vaccine against Neisseria gonorrhoeae;

·         Also missing is information on what proteins might be candidates for an effective vaccine.

Author Response

Thank you !

Round 2

Reviewer 1 Report

I would like to thanks to the authors for corrections and including important missing reference and improving writing.

I have no further comments

Reviewer 2 Report

I would advise authors to avoid rushing before submitting articles and double-check manuscripts.